# Geographic Location Encoding with Spherical Harmonics and Sinusoidal Representation Networks

**Marc Rußwurm**[*]
Wageningen University
Laboratory of Geo-information Science and Remote Sensing

**Konstantin Klemmer**
Microsoft Research New England

**Esther Rolf**
Harvard Data Science Initiative and
Center for Research on Computation and Society
University of Colorado, Boulder

**Robin Zbinden & Devis Tuia**
Environmental Computational Science and
Earth Observation Laboratory (ECEO)
École Polytechnique Fédérale de Lausanne (EPFL)

## Abstract

Learning representations of geographical space is vital for any machine learning model that integrates geolocated data, spanning application domains such as remote sensing, ecology, or epidemiology. Recent work embeds coordinates using sine and cosine projections based on Double Fourier Sphere (DFS) features. These embeddings assume a rectangular data domain even on global data, which can lead to artifacts, especially at the poles. At the same time, little attention has been paid to the exact design of the neural network architectures with which these functional embeddings are combined. This work proposes a novel location encoder for globally distributed geographic data that combines spherical harmonic basis functions, natively defined on spherical surfaces, with sinusoidal representation networks (SirenNets) that can be interpreted as learned Double Fourier Sphere embedding. We systematically evaluate positional embeddings and neural network architectures across various benchmarks and synthetic evaluation datasets. In contrast to previous approaches that require the combination of both positional encoding and neural networks to learn meaningful representations, we show that both spherical harmonics and sinusoidal representation networks are competitive on their own but set state-of-the-art performances across tasks when combined. The model code and experiments are available at https://github.com/marccoru/locationencoder.

## 1 Introduction

Location information is informative meta-data in many geospatial applications, ranging from species distribution modeling (Mac Aodha et al., 2019; Cole et al., 2023) and wildlife monitoring (Jeantet & Dufourq, 2023) to solar irradiance forecasting (Boussif et al., 2023), global tree height prediction (Lang et al., 2022) or the estimation of housing prices (Klemmer et al., 2023). Coordinates also serve as a task-descriptive feature for meta-learning across geographically distributed tasks (Tseng et al., 2022) or as input for geographic question answering (Mai et al., 2020a). Location encoders alone can be used to learn an implicit neural representation of a dataset (Stanley, 2007), for example, to compress weather data (Huang & Hoefler, 2023) or to learn an implicit neural representation of species distribution maps (Cole et al., 2023).

Methodologically, predictive modeling with spatial coordinates as inputs has a long history and is classically done with non-parametric approaches, like Gaussian Processes (GP), i.e., Kriging, (Oliver & Webster, 1990; Matheron, 1969) or kernel methods in general. For instance, Berg et al. (2014) proposed adaptive kernels for bird species recognition, where a class-specific kernel is learned for each species separately. Relying on spatially close training points is intuitive and

---

[*]corresponding author. Please email marc.russwurm@wur.nl

highly effective for interpolation tasks and small datasets. Still, it becomes computationally expensive when kernel similarities have to be computed between many test points and large training sets. And while research in scaling kernel methods to higher-dimensional data has progressed (Gardner et al., 2018), recent focus on scalable methods has shifted to using neural network weights to approximate the training points (Tang et al., 2015; Mac Aodha et al., 2019; Chu et al., 2019; Huang & Hoefler, 2023). Inspired by Double Fourier Sphere (DFS) (Orszag, 1974) and the success of positional encodings in transformers, a series of works developed and compared different sine-cosine embedding functions at different spatial scales (frequencies) where the number of scales controls the smoothness of the interpolation (Mai et al., 2020b; 2023b). Even though these DFS-inspired approaches have proven effective, they still assume a rectangular domain of longitude and latitude coordinates, which does not accurately reflect our planet's spherical geometry.

In this work, we instead propose to use spherical harmonic basis functions as positional embeddings that are well-defined across the entire globe, including the poles. In particular, the scientific use of location encoders requires good global performance, with poles being especially important for climate science. Methodologically, we build upon recent work here has been focused on learning functions on general manifolds (Koestler et al., 2022) and graphs (Grattarola & Vandergheynst, 2022) in non-euclidean geometries. This work focuses on spheres, a special case of non-euclidean geometries, where we utilize spherical harmonic basis functions, which are the eigenfunctions of the Laplace-Beltrami operator on the sphere (Koestler et al., 2022)

In summary, we

1. propose Spherical Harmonic (SH) coordinate embeddings that work well with all tested neural networks, and especially well when paired with Sinusoidal Representation Networks (SirenNet) (Sitzmann et al., 2020), which was untested for location encoding previously;

2. show by analytic comparison that SirenNets can be seen as learned Double Fourier Sphere embedding, which explains its good performance even with direct latitude/longitude inputs;

3. provide a comprehensive comparative study using the cross-product of positional embedding strategies and neural networks across diverse synthetic and real-world prediction tasks, including interpolating ERA5 climate data and species distribution modeling.

## 2 METHOD

### 2.1 LOCATION ENCODING

We encode a global geographic coordinate as longitude $\lambda \in [-\pi, \pi]$ and latitude $\phi \in [-\pi/2, \pi/2]$ to predict a target variable $y$, which can be a classification logit, a regression value, or a vector that can be integrated into another neural network, such as a vision encoder.

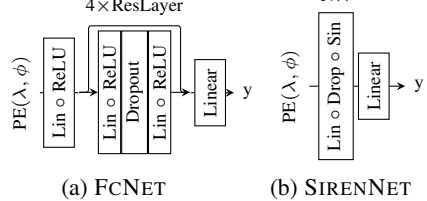

(a) FCNET    (b) SIRENNET

Figure 1: Overview over neural networks (SIRENNET Sitzmann et al. (2020) and FCNET Mac Aodha et al. (2019))

Following the convention in (Mai et al., 2022), these location encoding approaches generally take the form of

$$y = \text{NN}\left(\text{PE}\left(\lambda, \phi\right)\right) \qquad (1)$$

that combines a non-parametric positional embedding (PE) with a neural network (NN). Most previous work focuses on the positional encoding, typically linked together with a fully connected residual network (FCNET, see fig. 1a). This FCNET implementation with four residual layers (ResLayer) was originally proposed in Mac Aodha et al. (2019) for location encoding, and has been re-used as-is in follow-up work (Mai et al., 2023b; 2020b; Jeantet & Dufourq, 2023; Mai et al., 2023a; Cole et al., 2023). However, similar networks have been proposed earlier for pose estimation Martinez et al. (2017). Here, we discuss the main families of positional embedding approaches, deferring a detailed description to appendix A.2. We then describe our approach for a novel global positional encoder, linked with SIRENNET (fig. 1b), a network which is broadly used for learning implicit neural representations of images (Mildenhall et al., 2021) but has not been tested in this context of location encoding.

**DFS-based embeddings.** In their foundational study, Mac Aodha et al. (2019) use a simple WRAP positional encoding $\text{PE}(\lambda, \phi) = [\cos \lambda, \sin \lambda, \cos \phi, \sin \phi]$, which removed the discontinuity at the

LINEAR "Neural Network"

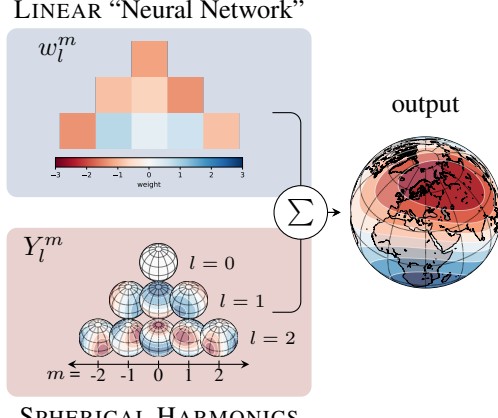

$w_l^m$

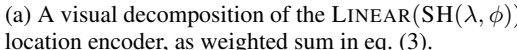

$$\text{weight}$$

$-3 \quad -2 \quad -1 \quad 0 \quad 1 \quad 2 \quad 3$

$\Sigma$

output

$Y_l^m$

$l = 0$

$l = 1$

$l = 2$

$m = -2 \quad -1 \quad 0 \quad 1 \quad 2$

SPHERICAL HARMONICS

(a) A visual decomposition of the LINEAR(SH$(\lambda, \phi)$) location encoder, as weighted sum in eq. (3).

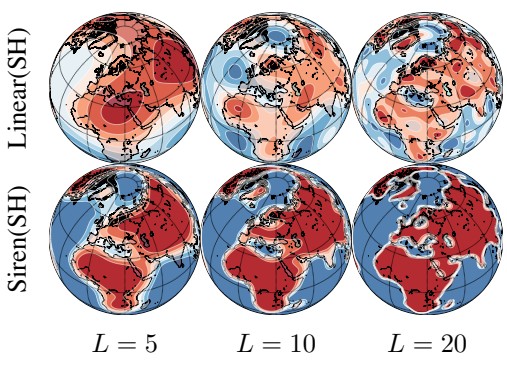

Linear(SH)

Siren(SH)

$L = 5 \qquad L = 10 \qquad L = 20$

(b) Left to right: increasing the number of Legendre polynomials $L$ increases the resolveable resolution by adding additional high frequency orders $m$ and degrees $l$. Using a more complex neural network, e.g., a 2-layer SIRENNET (bottom row), further increases the model's ability to resolve fine-grained detail.

Figure 2: SPHERICAL HARMONICS are orthogonal basis functions defined on spheres. Spherical functions can be naturally defined as the sum of parameter-weighting basis functions, as done in a linear layer (panel (a)). In this work, we propose a location encoder $NN(PE(\lambda, \phi))$ with a spherical harmonic positional embedding (PE) function with the SIRENNET neural network (NN), i.e., SIREN(SH), to learn complex functions defined on the globe, as illustrated in the example of land-ocean classification in the right panel.

dateline ($\lambda = \pi$). The original wrap positional encoding has since been identified as a special case of Double Fourier Sphere (DFS) (Orszag, 1974) coefficients by Mai et al. (2023b):

$$\text{DFS}(\lambda, \phi) = \bigcup_{n=0}^{S-1} [\sin \phi_n, \cos \phi_n] \cup \bigcup_{m=0}^{S-1} [\sin \lambda_m, \cos \lambda_m] \cup$$
$$\bigcup_{n=0}^{S-1} \bigcup_{m=0}^{S-1} [\cos \phi_n \cos \lambda_m, \cos \phi_n \sin \lambda_m, \sin \phi_n \cos \lambda_m, \sin \phi_m \sin \lambda_m] \tag{2}$$

where the coordinates $\phi, \lambda$ are introduced across multiple frequencies $\lambda_s = \frac{\lambda}{\alpha_s}$ and $\phi_s = \frac{\phi}{\alpha_s}$ by a scale-dependent factor $\alpha_s = r_{\min} \cdot \left(\frac{r_{\max}}{r_{\min}}\right)^{\frac{s}{S-1}}$ that allows users to specify the minimum and maximum radii $r_{\min}, r_{\max}$ that can be resolved at multiple scales up to $S$. The union operator $\bigcup$ indicates the concatenation of these functions into one large vector.

The full DFS embedding is computationally expensive and quantitatively less accurate than the special cases SPHEREM and SPHEREC (Mai et al., 2023b). Hence, Mai et al. (2023b) proposed SPHEREM+ and SPHEREC+, which is a combination with the GRID embedding function of Mai et al. (2020b). In the same paper, Mai et al. (2020b) proposed to use hexagonal basis functions rather than rectangular grid cells, which is the THEORY embedding function that we will compare to in the results section.

**Baseline embeddings.** DIRECT, i.e., using the identity function to encode $(\phi, \lambda)$, or CARTESIAN3D, which correponds to encoding $(\phi, \lambda)$ into $(x, y, z)$ coordinates, are also common baselines still used in recent work (Tseng et al., 2022).

**Our Approach.** We depart from DFS and instead propose to use SPHERICAL HARMONIC (SH) basis functions (detailed in section 2.2), which have a long tradition in the Earth sciences for representing the gravity field (Pail et al., 2011) or the magnetosphere of planets (Smith et al., 1980). The SH basis functions can be multiplied with learnable coefficient weights. Examples of SH basis functions and their corresponding learnable triangle weight matrix are shown in fig. 2a. The weight matrix can be expressed as a linear layer, i.e., a very simple "neural network" following the location encoding terminology of eq. (1). It can also be combined with more complex neural networks; we propose to use Sinusoidal Representation Networks (SIRENNET) (Sitzmann et al., 2020)–shown in fig. 1b–which are architecturally simpler than the FCNET (fig. 1a) used in related works. We compare SIRENNET with DFS encodings used in previous location encoding approaches in section 2.3.

## 2.2 WEIGHTED SPHERICAL HARMONICS

Any function on a sphere

$$f(\lambda, \phi) = \sum_{l=0}^{\infty} \sum_{m=-l}^{l} w_l^m Y_l^m(\lambda, \phi) \tag{3}$$

with surface points defined by $(lat, lon) = (\phi, \lambda)$ can be expressed by the $w_l^m$ weighted sum of orthogonal spherical harmonic basis functions $Y_l^m$ of increasingly higher-frequency degrees $l$ and orders $m$. In practice, we choose a maximum number of Legendre polynomials $L$ (instead of $\infty$ in eq. (3)) to approximate a spherical function up to the highest-frequency harmonic. To illustrate, we draw examples of harmonics up to $L = 3$ in fig. 2a (bottom left) alongside their coefficients (top left), which combined approximate the landmass outline of our planet (right) with only nine trainable weights. Each individual harmonic

$$Y_l^m(\lambda, \phi) = \sqrt{\frac{2l+1}{4\pi} \frac{(l-|m|)!}{(l+|m|)!}} P_l^m(\cos \lambda) e^{im\phi} \tag{4}$$

can be calculated with an *associated Legendre polynomial* $P_l^m$. This polynomial can be analytically constructed by repeated differentiation $P_l^m(x) = (-1)^m (1-x^2)^{\frac{m}{2}} \frac{d^m}{dx^m}(P_l(x))$ of their zero-order ($m = 0$) Legendre polynomials $P_l = P_l^0$. Zero-order polynomials of a certain degree $l$ are similarly obtained by repeated differentiation $P_l(x) = \frac{1}{2^l l!} \frac{d^l}{dx^l} \left[ (x^2 - 1)^l \right]$. To obtain associated polynomials of negative order ($m < 0$), symmetries $P_l^{-m}(x) = (-1)^m \frac{(l-m)!}{(l+m)!} P_l^m(x)$ can be used.

We are interested in the real form of the original complex eq. (4), which can be implemented as

$$Y_{l,m}(\lambda, \phi) = \begin{cases} (-1)^m \sqrt{2} \bar{P}_l^{|m|}(\cos \lambda) \sin(|m|\phi), & \text{if } m < 0 \\ \bar{P}_l^m(\cos \lambda), & \text{if } m = 0 \\ (-1)^m \sqrt{2} \bar{P}_l^m(\cos \lambda) \cos(m\phi), & \text{if } m > 0 \end{cases} \tag{5}$$

with cosine and sine functions, where $\bar{P}_l^m(\cos \lambda) = \sqrt{\frac{2l+1}{4\pi} \frac{(l-|m|)!}{(l+|m|)!}} P_l^m(\cos \lambda)$ denotes the normalized associated Legendre polynomial. The polynomials can be computed iteratively in closed-form $P_l^m(x) = (-1)^m \cdot 2^l \cdot (1-x^1)^{\frac{m}{2}} \cdot \sum_{k=m}^{l} \frac{k!}{(k-m!)} \cdot x^{k-m} \cdot \binom{N}{k} \cdot \binom{\frac{l+k-1}{2}}{l}$ following Green (2003). However, pre-computing the polynomials analytically is more computationally efficient than using the closed form, as we show empirically later in the results section. We use this analytically pre-computed implementation in experiments throughout the paper.

## 2.3 SINUSOIDAL REPRESENTATION NETWORKS AS LEARNED DFS EMBEDDINGS

Besides SPHERICAL HARMONICS, we propose to use Sinusoidal Representation Networks (SIRENNETS) (Sitzmann et al., 2020) as a preferred neural network for location encoding. We now show that SIRENNETS can be seen as a form of learned Double Fourier Sphere (DFS) positional embedding.

We use the GRID embedding function by Mai et al. (2023b) as a simple example, but the following comparison can also be made for other DFS embeddings. GRID is defined as: $\text{GRID}(\lambda, \phi) = \bigcup_{s=0}^{S-1} [\cos \lambda_s, \sin \lambda_s, \cos \phi_s, \sin \phi_s] = \bigcup_{s=0}^{S-1} \left[ \cos(\frac{\lambda}{\alpha_s}), \sin(\frac{\lambda}{\alpha_s}), \cos(\frac{\phi}{\alpha_s}), \sin(\frac{\phi}{\alpha_s}) \right]$, where $\alpha_s$ is a scaling factor that induces increasingly high frequencies through scales $s$ from 0 to $S - 1$ and $\bigcup$ indicates concatenation. With the 90-degree phase shift between sine and cosine, i.e., $\cos(\theta) = \sin(\theta + \frac{\pi}{2})$, we can rewrite GRID solely using sine functions

$$\text{GRID}(\lambda, \phi) = \bigcup_{s=0}^{S-1} \left[ \sin(\frac{\lambda}{\alpha_s} + \frac{\pi}{2}), \sin(\frac{\lambda}{\alpha_s}), \sin(\frac{\phi}{\alpha_s} + \frac{\pi}{2}), \sin(\frac{\phi}{\alpha_s}) \right], \tag{6}$$

where every block of four feature dimensions (indexed by $s$) is scaled by the same constant factor $\frac{1}{\alpha_s}$ and every odd feature embedding is phase shifted ($+\frac{\pi}{2}$).

Similarly, a simple 1-layer SIRENNET can be written out as $\text{SIRENNET}(\phi, \lambda) = \sin(\mathbf{W}[\phi, \lambda]^T + \mathbf{b}) = \sin(\mathbf{w}^\phi \phi + \mathbf{w}^\lambda \lambda + \mathbf{b})$ with $\mathbf{W} = [\mathbf{w}^\lambda, \mathbf{w}^\phi] \in \mathbb{R}^{[H,2]}$, $\mathbf{b} \in \mathbb{R}^H$ trainable weights and $H$ hidden output dimensions. Writing out the vector-matrix multiplication as concatenated scalars results in $\text{SIRENNET}(\lambda, \phi) = \bigcup_{h=1}^{H}[\sin(w_h^\lambda \lambda + w_h^\phi \phi + b)]$. If we now set every $w_h^\phi = w_{h+1}^\phi = 0$ and every $w_{h+2}^\lambda = w_{h+3}^\lambda = 0$, we similarly alternate between longitudinal and latitudinal scaling for each feature. If we now also set $b_{h+1} = b_{h+3} = 0$, we arrive at a GRID-like SIREN variant

$$\text{GRIDSIREN}(\lambda, \phi) = \bigcup_{h=0,4,\ldots}^{H-1} [\sin(w_h^\lambda \lambda + b_h), \sin(w_{h+1}^\lambda \lambda), \sin(w_{h+2}^\phi \phi + b_{h+2}), \sin(w_{h+3}^\phi \phi)], \quad (7)$$

which corresponds to eq. (6) if $b_h = b_{h+2} = \frac{\pi}{2}$ and all $w_h^\lambda = w_{h+1}^\lambda = w_{h+2}^\phi = w_{h+3}^\phi = \frac{1}{\alpha_s}$. Hence, a 1-layer SIRENNET with these hard-coded weights equals the GRID DFS encoding, where the longitudes and latitudes are manually scaled to produce lower or higher frequencies in each embedding dimension. Crucially, these scaling factors are learned in GRIDSIREN through gradient descent, while they are fixed for every block of four in the GRID embedding in eq. (6). This observation highlights that SIRENNET blurs the distinction between Positional Embedding (PE) and Neural Network (NN) from eq. (1) and analytically explains the good performance of SIRENNET even with a simple DIRECT embedding functions, as we will show later.

## 3    DATASETS AND EXPERIMENTS

The experiments in this work span from controllable synthetic problems to real-world data. Across all datasets, the location encoder model is fitted on training data to learn an underlying signal defined by the training label (probability map or regression value). How well the signal has been approximated in the network weights is measured on spatially different test points.

### 3.1    CHECKERBOARD CLASSIFICATION: A CONTROLLABLE SYNTHETIC TOY DATASET

A Fibonacci lattice is a mathematical idealization of natural patterns on a sphere with optimal packing, where the surface area represented by each point is almost identical (González, 2010) within an error of 2% (Swinbank & James Purser, 2006). It is calculated for 2N points ($i$ ranges from $-N$ to $N$) as $\phi_i = \arcsin\frac{2i}{2N+1}$ and $\lambda_i = 2\pi i \Phi$, with the Golden Ratio $\Phi = \frac{1}{2}(1+\sqrt{5}) \approx 1.618$. To make this a classification problem, we choose 100 points according to the Fibonacci-lattice and assign each point one of 16 classes in a regular order ($i$ modulo 16). In later experiments, we increase or decrease the number of grid cells to vary the spatial scale and measure the maximum resolvable resolution of location encoders. For the test dataset (fig. 3a), we sample a second Fibonacci-lattice grid with 10 000 points and assign each point the class of the closest label point using the spherical Haversine distance. For training and validation sets, we uniformly sample 10 000 points on the sphere and similarly assign the label of the closest labeled point.

### 3.2    LAND-OCEAN CLASSIFICATION: IMPLICIT NEURAL REPRESENTATION OF LAND

We also design a more realistic land-ocean classification dataset shown in fig. 3b. This dataset tests the encoder's ability to learn patterns across different scales and resolutions, while providing training and evaluation points across the entire globe. For training and validation data, we sample 5000 points uniformly on the sphere's surface and assign a positive label for land and a negative label for water depending on whether they are within landmasses of the "Natural Earth Low Resolution" shapefile[1]. For the test dataset, we generate 5000-equally spaced points using a Fibonacci-lattice.

### 3.3    MULTI-VARIABLE ERA5 INTERPOLATION

Next, we use real-world climate data. We downloaded globally distributed climate variables at Jan. 1, 2018, at 23:00 UTC from the fifth-generation atmospheric reanalysis of the global climate (ERA5) product of the European Centre for Medium-Range Weather Forecasts. This dataset contains 6 483 600 observations corresponding to regularly gridded locations on the surface of our

---

[1]https://www.naturalearthdata.com/http//www.naturalearthdata.com/download/50m/physical/ne_50m_land.zip

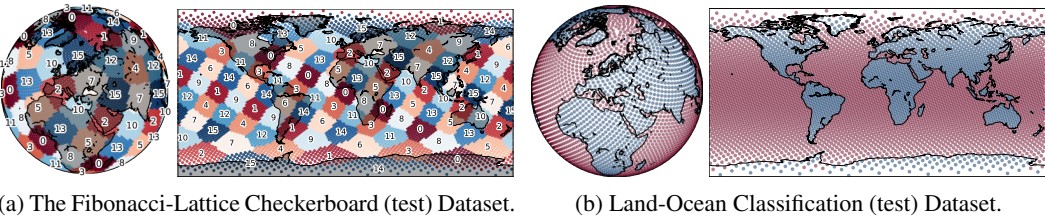

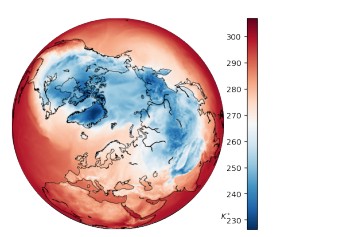

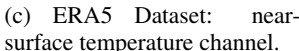

(a) The Fibonacci-Lattice Checkerboard (test) Dataset.

(b) Land-Ocean Classification (test) Dataset.

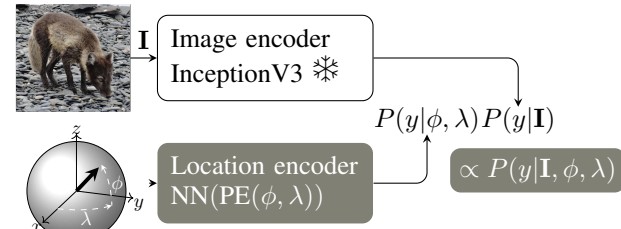

(c) ERA5 Dataset: near-surface temperature channel.

(d) iNaturalist 2018 setup: We vary the location encoder and measure classification accuracy increase thanks to location $\phi, \lambda$.

Figure 3: Datasets in this work from synthetic (a) and semi-synthetic (b) to real use-cases (c,d).

planet. At each location, we obtain eight continuous climate variables: the $u$ and $v$ components of near-surface wind speed, near-surface air temperature, surface air pressure, near-surface specific humidity, surface shortwave and longwave radiation, and total precipitation (rainfall and snowfall flux). An illustration of near surface air temperature (centered on the North pole) is shown in fig. 3c. We design a multi-task regression in which the model learns all eight variables jointly from a small (1%) randomly sampled training set. Another 5% of the data is reserved for the validation set. Through this dataset, we approximate the continuous signal underlying the gridded climate data into the model weights of a neural network and measure its ability to reproduce the signal continuously at any point without explicit interpolation of neighboring grid cells.

### 3.4 iNATURALIST 2018: LOCATION-INFORMED IMAGE CLASSIFICATION

Finally, we test the location encoders' ability to encode location as auxiliary information for fine-grained species classification with georeferenced images. Here, we use the iNaturalist (iNat2018) (Van Horn et al., 2018) dataset, which contains georeferenced images of plant and animal species. It is a real-world dataset with several data-inherent challenges: it presents a fine-grained classification problem with 8142 individual categories in a long-tailed class distribution. It has a strong sampling bias toward the US and Europe due to crowd-sourcing. Adding location information to an image classifier has been proven beneficial, as, e.g., similarly-looking species from different continents can be ruled out. Hence, iNaturalist 2018 has been used extensively to benchmark location encoders (Mac Aodha et al., 2019; Mai et al., 2020b; 2023a). Following these works, we used the public training split of 436 063 georeferenced images and partitioned it randomly into training and validation datasets in an 80:20 ratio. The public validation split of 24 343 georeferenced images was used as a test dataset. We combine the probabilities of the location encoder $P(y|\phi, \lambda)$ and image classifier $P(y|\mathbf{I})$ by multiplication, as shown schematically in fig. 3d. A fixed pre-trained InceptionV3 network from Mac Aodha et al. (2019) serves as the image classifier. The results section reports the relative accuracy increase that the location encoder adds to the base image classifier. Through this dataset, we learn a continuous species distribution map in the model weights from point observations of training points. At test time, the model predicts the probability of species presence at new test locations, which informs a separate image classifier and increases the overall prediction accuracy.

## 4 RESULTS

The implementation details and hyperparameter tuning protocol can be found in appendix A.1.

### 4.1 COMPARATIVE PERFORMANCE

**Checkerboard and Land-Ocean Classification**. tables 1a and 1b show averaged classification accuracies and standard deviations over five runs of the checkerboard and land-ocean classification datasets, respectively. We observe that the SPHERICAL HARMONICS (SH) embedding works well

Table 1: Comparison of different positional embeddings (rows) and neural networks (columns). The best combination is bold, and the best positional embedding per neural network is underlined.

(a) Checkerboard Results: Classification accuracy and standard deviation from 5 runs; higher is better.

| PE ↓    NN → | LINEAR | FCNET | SIRENNET |
|---|---|---|---|
| DIRECT | $8.1 \pm 0.3$ | $30.0 \pm 11.2$ | $94.7 \pm 0.1$ |
| CARTESIAN3D | $8.5 \pm 0.4$ | $85.5 \pm 1.2$ | $95.7 \pm 0.2$ |
| WRAP | $10.2 \pm 0.4$ | $68.3 \pm 11.5$ | $95.3 \pm 0.1$ |
| GRID | $10.5 \pm 0.6$ | $93.5 \pm 0.3$ | $93.7 \pm 0.4$ |
| THEORY | $64.0 \pm 0.1$ | $93.4 \pm 0.3$ | $94.2 \pm 0.3$ |
| SPHEREC | $15.4 \pm 0.5$ | $93.3 \pm 0.3$ | $93.9 \pm 0.2$ |
| SPHEREC+ | $23.5 \pm 0.6$ | $93.7 \pm 0.4$ | $94.4 \pm 0.1$ |
| SPHEREM | $6.6 \pm 0.3$ | $78.3 \pm 0.9$ | $68.0 \pm 0.8$ |
| SPHEREM+ | $10.8 \pm 0.2$ | $87.9 \pm 0.3$ | $90.0 \pm 0.2$ |
| SH (ours) | $\underline{94.8 \pm 0.1}$ | $\underline{94.9 \pm 0.1}$ | $\mathbf{95.8 \pm 0.0}$ |

(b) Land-Ocean Results: Classification accuracy and standard deviation from 5 runs; higher is better.

| PE ↓    NN → | LINEAR | FCNET | SIRENNET |
|---|---|---|---|
| DIRECT | $71.4 \pm 0.0$ | $90.3 \pm 0.7$ | $95.1 \pm 0.3$ |
| CARTESIAN3D | $70.5 \pm 3.5$ | $92.7 \pm 0.3$ | $92.8 \pm 0.3$ |
| WRAP | $74.4 \pm 0.3$ | $93.2 \pm 0.3$ | $95.2 \pm 0.2$ |
| GRID | $81.7 \pm 0.1$ | $95.1 \pm 0.1$ | $95.5 \pm 0.2$ |
| THEORY | $86.9 \pm 0.1$ | $94.9 \pm 0.2$ | $95.5 \pm 0.1$ |
| SPHEREC | $79.6 \pm 0.2$ | $95.0 \pm 0.3$ | $95.2 \pm 0.1$ |
| SPHEREC+ | $84.6 \pm 0.2$ | $95.3 \pm 0.1$ | $95.5 \pm 0.1$ |
| SPHEREM | $74.0 \pm 0.0$ | $89.1 \pm 0.1$ | $88.3 \pm 0.4$ |
| SPHEREM+ | $81.9 \pm 0.2$ | $92.1 \pm 0.3$ | $93.7 \pm 0.1$ |
| SH (ours) | $\underline{94.4 \pm 0.1}$ | $\mathbf{95.9 \pm 0.1}$ | $\underline{95.8 \pm 0.1}$ |

(c) Inaturalist 2018: Improved classification accuracy with location. Standard dev. from 5 runs; higher is better.

| PE ↓    NN → | LINEAR | FCNET | SIRENNET |
|---|---|---|---|
| DIRECT | $-5.9 \pm 0.1$ | $+9.3 \pm 0.3$ | $+12.1 \pm 0.1$ |
| CARTESIAN3D | $+0.8 \pm 0.2$ | $+11.8 \pm 0.1$ | $+12.0 \pm 0.1$ |
| WRAP | $-0.1 \pm 0.1$ | $\underline{+12.1 \pm 0.1}$ | $+12.1 \pm 0.1$ |
| GRID | $+11.2 \pm 0.1$ | $+11.8 \pm 0.2$ | $+11.6 \pm 0.4$ |
| THEORY | $+11.5 \pm 0.0$ | $+10.8 \pm 0.0$ | $+11.4 \pm 0.1$ |
| SPHEREC | $+11.2 \pm 0.2$ | $+12.0 \pm 0.2$ | $+\mathbf{12.3 \pm 0.1}$ |
| SPHEREC+ | $+11.1 \pm 0.2$ | $+11.5 \pm 0.3$ | $+10.3 \pm 0.4$ |
| SPHEREM | $+7.2 \pm 0.2$ | $+11.3 \pm 0.2$ | $+10.6 \pm 0.6$ |
| SPHEREM+ | $\underline{+11.6 \pm 0.1}$ | $+12.0 \pm 0.1$ | $+10.7 \pm 0.2$ |
| SH (ours) | $\underline{+10.5 \pm 0.1}$ | $+12.0 \pm 0.0$ | $+\mathbf{12.3 \pm 0.2}$ |

| PE ↓    NN → | LINEAR | FCNET | SIRENNET |
|---|---|---|---|
| DIRECT | $-4.7 \pm 0.1$ | $+7.1 \pm 0.2$ | $+8.8 \pm 0.1$ |
| CARTESIAN3D | $+0.8 \pm 0.1$ | $\underline{+8.6 \pm 0.1}$ | $+8.7 \pm 0.1$ |
| WRAP | $-0.1 \pm 0.1$ | $\underline{+8.6 \pm 0.2}$ | $+8.7 \pm 0.1$ |
| GRID | $+8.1 \pm 0.1$ | $+8.3 \pm 0.1$ | $+8.5 \pm 0.1$ |
| THEORY | $\underline{+8.2 \pm 0.1}$ | $+8.0 \pm 0.1$ | $+8.3 \pm 0.2$ |
| SPHEREC | $+7.9 \pm 0.1$ | $+8.3 \pm 0.1$ | $+8.8 \pm 0.1$ |
| SPHEREC+ | $+7.8 \pm 0.0$ | $+8.3 \pm 0.1$ | $+7.8 \pm 0.2$ |
| SPHEREM | $+4.9 \pm 0.1$ | $+7.5 \pm 0.2$ | $+7.8 \pm 0.3$ |
| SPHEREM+ | $\underline{+8.2 \pm 0.1}$ | $+8.4 \pm 0.1$ | $+8.0 \pm 0.1$ |
| SH (ours) | $+7.5 \pm 0.1$ | $+8.4 \pm 0.1$ | $+\mathbf{9.0 \pm 0.1}$ |

| image-only: 59.2% top-1 accuracy with encoder NN(PE) ↑ | image-only: 77.0% top-3 accuracy with encoder NN(PE) ↑ |
|---|---|

for all neural networks, including the minimalist LINEAR layer. Similarly, SIRENNET (Sitzmann et al., 2020) achieves high accuracies consistently across all positional embeddings, while the FC-NET (Mac Aodha et al., 2019) shows larger variability, especially with simpler embedding functions, such as CARTESIAN3D, DIRECT, or WRAP.

**ERA5**. Table 2 shows the mean squared error averaged over the eight climate variables in the ERA5 dataset. Means and standard deviations are obtained from 10 runs with different random initializations. This multi-task regression is especially difficult as it requires the location encoder to learn the spatial patterns of each channel separately and their interactions. SPHERI-CAL HARMONIC embeddings consistently achieve the lowest error for all neural networks, while the combination with FC-NET was best with $0.58 \pm 0.02$. This constitutes a substantial performance gain over the next best existing competitor (SPHEREC+ and FCNET) with $1.38 \pm 0.03$.

Table 2: ERA 5 dataset: Interpolation of 8 climate variables. Averaged MSE and std. dev. from 10 runs; lower is better.

| PE ↓    NN → | LINEAR | FCNET | SIRENNET |
|---|---|---|---|
| DIRECT | $27.19 \pm 0.08$ | $7.83 \pm 1.06$ | $1.62 \pm 0.10$ |
| CARTESIAN3D | $24.18 \pm 0.00$ | $4.23 \pm 0.25$ | $1.57 \pm 0.11$ |
| WRAP | $13.26 \pm 0.02$ | $4.13 \pm 0.25$ | $1.89 \pm 0.07$ |
| GRID | $9.83 \pm 0.01$ | $1.51 \pm 0.04$ | $2.37 \pm 0.13$ |
| THEORY | $9.24 \pm 0.01$ | $1.61 \pm 0.05$ | $2.99 \pm 0.10$ |
| SPHEREC | $20.03 \pm 0.02$ | $1.92 \pm 0.06$ | $1.95 \pm 0.09$ |
| SPHEREC+ | $8.50 \pm 0.02$ | $1.38 \pm 0.03$ | $1.97 \pm 0.08$ |
| SPHEREM | $26.68 \pm 0.13$ | $3.51 \pm 0.08$ | $5.89 \pm 0.68$ |
| SPHEREM+ | $9.94 \pm 0.02$ | $1.54 \pm 0.07$ | $2.75 \pm 0.09$ |
| SH (ours) | $\underline{1.39 \pm 0.02}$ | $\mathbf{0.58 \pm 0.02}$ | $\underline{1.19 \pm 0.04}$ |

**iNaturalist**. Table 1c shows the relative top-1 and top-3 accuracy improvements that location information through $P(y|\mathbf{I}, \phi, \lambda)$ adds to an image-only classifier $P(y|\mathbf{I})$. The underlying image-only classification top-1 and top-3 accuracies are 59.2% and 77.0%, respectively. All location encoders improve the top-1 accuracy substantially between +9.3% (FCNET(DIRECT)) and +12.2% (SIRENNET(SH)) except for the trivial LINEAR(DIRECT) and too simple LINEAR(WRAP) encodings. At top-1 accuracy, the SPHERICAL HARMONIC embedding used with the SIRENNET network is slightly more accurate than the other approaches but still roughly within the standard deviations (between 0.1% and 0.4%) over five evaluation runs. However, for other neural networks, other positional encoders also achieve the best column score (underlined). At top-3 accuracy, we see lower variances in general between standard deviations of 0.1 percentage. Also here, SIREN(SH) is slightly better with $+9.0 \pm 0.1$ % compared to $+8.8 \pm 0.1$% of SIREN(DIRECT) and significantly better than SPHEREM+ (Mai et al., 2023b) with +8.4%.

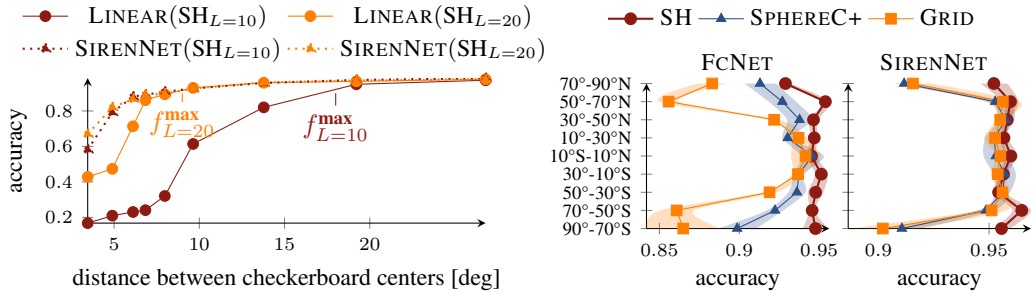

(a) Resolvable resolution: Accuracy at increasingly dense checkerboard patterns. Shorter distances (further left) require resolving higher-resolution signals.

(b) Latitudinal accuracy: Checkerboard accuracy in 20° bands from North to South for FcNET (left) and SIRENNET (right). Standard deviations from 5 runs.

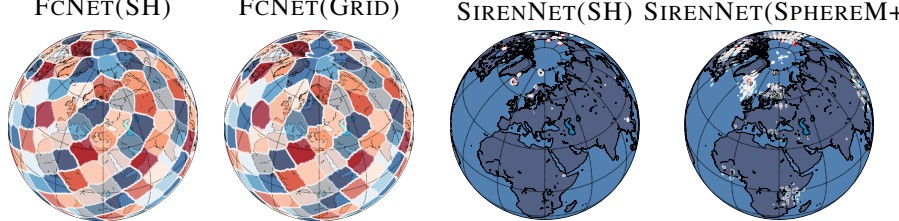

(c) Predictions on a regular grid with the Checkerboard (left two) and iNaturalist datasets (right two). The location encoders learn an implicit neural representation of the checkerboard patterns (left two) and the distribution map of Arctic Foxes $P(y = \text{arctic fox}|\lambda, \phi)$. Note the artifacts at North poles for the SPHEREM+ and GRID location embeddings that are not present with SPHERICAL HARMONICS (SH).

Figure 4: Experimental evaluation of properties of SPHERICAL HARMONICS paired with SIREN-NET. Fig a) demonstrates its ability to resolve signals across different resolutions. Figures b) and c) highlight the ability to learn signals close to the poles at high latitudes quantitatively (b) and qualitatively (c) on Checkerboard and iNaturalist data.

**Takeaways**. The SPHERICAL HARMONIC (SH) embedding function achieved the best results across all four datasets and consistently across different neural networks for three of them. For iNaturalist, SPHERICAL HARMONICS with SIRENNET was still best (bolded), but several other positional embeddings were better in combination with FcNET and LINEAR (underlined). We generally connect this to the location bias in iNaturalist, where only a few species observations are recorded at higher latitudes, where good representations of the spherical geometry are most advantageous. Comparing neural networks: the commonly used FcNET (Mac Aodha et al., 2019) achieved the best results in predicting ERA5 interpolation and Land-Ocean classification, while the SIRENNET network was best in the iNaturalist and checkerboard problems. Overall, SIRENNET performed well across all positional embedding functions, including the trivial DIRECT embedding. Similarly, SPHERICAL HARMONICS performed reasonably well, even with a simple LINEAR layer on all datasets.

## 4.2 PROPERTIES AND CHARACTERISTICS

Next, we investigate characteristics of SPHERICAL HARMONICS with SIRENNET with respect to key properties of location encoders: resolvable resolution, accuracy across latitudes, and computational efficiency.

**Resolvable Resolution**. In fig. 4a, we focus on resolution limitations of SPHERICAL HARMONICS (SH) of different orders $L$. On the checkerboard dataset, we increase the number of grid cells in the Fibonacci lattice, which produces denser and increasingly fragmented patterns. We report the average distance between points on the x-Axis (in ° degree) as an indication of resolution. We test two SPHERICAL HARMONIC positional embeddings with 10 and 20 Legendre polynomials ($SH_{L=10}, SH_{L=20}$). We primarily use a LINEAR layer as a "neural network" to highlight the resolution limitations of spherical harmonics alone, as it represents eq. (3) where a single trainable parameter weights one harmonic. In this case, we can analytically calculate the maximum frequency $f^{\text{max}} = \frac{180}{2L}$ that a linear spherical harmonic embedding should be able to resolve. In the plot, we observe a decrease in accuracy at resolutions higher than 9- and 18-degree resolutions, respectively. Hence, more Legendre polynomials $L$ allow the SH(LINEAR) model to resolve higher-frequency

components. Replacing the LINEAR layer with a more complex SIRENNET network (dotted lines) further improves the model's resolution, indicated by a higher accuracy at smaller distances between centers. Both SIRENNET(SH$_{L=20}$) and SIRENNET(SH$_{L=20}$) remain accurate up to grid cells spaced 6° degrees apart with a slight edge for SIRENNET(SH$_{L=20}$) thanks to the larger number of spherical harmonics. Hence, both increasing the number of Legendre polynomials and the complexity of the neural network (i.e., SIRENNET instead of LINEAR) helps resolve high-frequency signals on the sphere. This property can also be seen qualitatively in fig. 2b on the example of land-ocean classification.

**Latitudinal Accuracy**. In fig. 4b, we compare the checkerboard classification accuracies of SPHERICAL HARMONICS (SH), GRID (Mai et al., 2020b) and SPHEREC+ (Mai et al., 2023b) positional embeddings, combined with the FCNET (left) and SIRENNET (right) networks separated by 20° north-south spanning latitude bands. The DFS-based encodings (GRID and SPHEREC+) assume a rectangular domain of latitude and longitude. This assumption is violated at the poles (90° North and 90° South) where all meridians converge to a single point. This leads to polar prediction artifacts in the checkerboard pattern, as shown in fig. 4c (left), and consequently to lower accuracies at higher latitudes. The SIRENNET network helps reduce these artifacts at the 50° to 70° bands N & S, but a performance degradation is still clearly observable close to the poles (70°-90°). In contrast, no substantial degradation at higher latitudes is visible on the location encoder with the spherical harmonic embedding function with both FCNET (left) and SIRENNET (right) networks, as spherical harmonics are well-defined on the entire sphere. Such artifacts also occur on real-world datasets like iNaturalist, as shown as vertical lines along the North-South meridians on the learned Arctic Fox prediction map from iNaturalist shown in fig. 4c (right) with the FCNET(GRID) location encoder.

**Computational Efficiency of Positional Encoders**.
Figure 5 shows the computational impact measured in inference runtime of 10 000 test points of the checkerboard dataset with an increasing number of Legendre polynomials (or scales for SPHEREC+) on the checkerboard dataset. We first compare a spherical harmonics implementation that uses the closed form implementation (SH-closed-form) following (Green, 2003) with the SPHEREC+ embedding. The closed-form implementation (SH-closed-form) scales poorly with large orders, as the inference time reaches 10 seconds with $L = 40$ compared to 0.3s of SPHEREC+ (Mai et al., 2023b). To overcome this limitation, we calculate the individual spherical harmonics analyti-

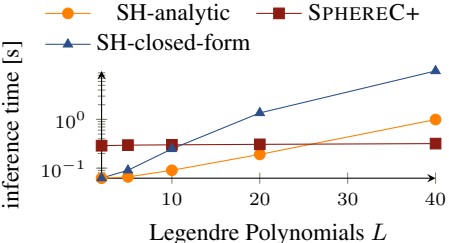

Figure 5: computational efficiency of analytic and pre-computed SH implementations with SphereC+ on the checkerboard dataset.

cally for each harmonic with eq. (5) and simplify the equation with SymPy (Meurer et al., 2017). This leads to a substantial performance improvement at 1 second for $L = 40$ over the closed-form implementation. All other experimental results in the paper use this analytic calculation, which is faster regarding runtime than the provided SPHEREC+ implementation up to $L > 20$ polynomials.

## 5 CONCLUSION AND OUTLOOK

Encoding geographical locations effectively is key to meeting many real-world modeling needs, and it requires considering our planet's spherical geometry. To this end, we propose to use orthogonal SPHERICAL HARMONIC basis functions paired with sinusoidal representation networks (SIRENNETs) to learn representations of geographic location. Even though these spherical harmonics are not as trivial to implement as classical sine-cosine functions, they are inherently better suited to represent functions on the sphere. They can be applied at a competitive computational runtime to existing sine-cosine encodings. Regarding neural networks, we have shown that SIRENNET can be seen as a learned Double Fourier Sphere positional embedding, which provides an intuition to explain its good performance even with a simple DIRECT positional embedding. This blurs the established division of positional embedding and neural networks established in previous work (Mai et al., 2022) and opens new research questions towards learning more parameter-efficient positioned embeddings. We hope to have provided insight into best practices for location encoding of large-scale geographic data and can concretely recommend SIRENNET for any problem involving geographic coordinates and SPHERICAL HARMONIC embeddings for problems that involve data at a global scale or in polar regions, which are not represented well by existing approaches.

## 6 ACKNOWLEDGEMENTS

RZ and DT acknowledge financial support from the Swiss National Science Foundation on project DeepHSM (n. 200021_204057). ER acknowledges support from the Harvard Data Science Initiative and the Center for Research on Computation and Society.

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

# A  SUPPLEMENTARY INFORMATION

## A.1  IMPLEMENTATION DETAILS AND HYPERPARAMETER TUNING

We use the same protocol across all experiments. To find optimal hyperparameters, we minimize validation loss in 100 iterations (30 iterations for iNaturalist 2018) with the Optuna framework (Akiba et al., 2019) in combination with Pytorch Lightning (Falcon & The PyTorch Lightning team, 2019) for each positional embedding and neural network combination separately. For the DFS-based positional embeddings (GRID, THEORY, SPHEREC, SPHEREC+, SPHEREM, and SPHEREM+), we tune the minimum radius $r_{min}$ between 1 and 90 degrees in 9-degree steps. We keep the maximum radius $r_{max}$ fixed at 360 degrees, as all problems have global or continental scale (sea-ice thickness) scale. We further tune the number of frequencies $S$ between 16 and 64 with steps of size 16 (32 for ERA5). For the spherical harmonics embeddings, we tune the number of Legendre polynomials $L$ between 10 and 30 in steps of 5 polynomials. In terms of neural networks, we vary the number of hidden dimensions between 32 and 128 in 32-dimension steps both for SIREN (Sitzmann et al., 2020) and FCNET (Mac Aodha et al., 2019) and vary the number of layers between one and three for Siren. For all combinations, we tune the learning rate on a logarithmic scale between $10^{-4}$ and $10^{-1}$ and the weight decay between $10^{-8}$ and $10^{-1}$.

Annotations provided in the iNaturalist 2018 dataset are presence-only, which means that the presence of a species is not informative of the absence of another. This makes the commonly used cross-entropy loss sub-optimal, as the classes are not mutually exclusive. Instead, we use the full "assume negative" loss function proposed recently by (Cole et al., 2023), defined as

$$\mathcal{L}(\boldsymbol{y}, \hat{\boldsymbol{y}}) = -\frac{1}{S} \sum_{s=1}^{S} \left[ 1_{[y_s=1]} \lambda \log(\hat{y}_s) + 1_{[y_s \neq 1]} \log(1 - \hat{y}_s) + \log(1 - \hat{y}_s') \right], \quad (8)$$

where $S = 8142$ is the number of species, $\lambda$ weighs the positive term of the loss, and $\hat{\boldsymbol{y}}'$ is a prediction at a randomly sampled location across the globe.

During fitting/training the model, we monitor the validation loss and perform early stopping with a patience of 30 epochs (10 epochs for iNaturalist) if the loss has not decreased. We do this for all NN-PE combinations consistently.

## A.2  COMPARISON METHODS: LOCATION ENCODINGS

All encoding approaches take the form of

$$y = \text{NN}\left(\text{PE}\left(\lambda, \phi\right)\right) \quad (9)$$

where a non-parametric positional embedding (PE) projects the raw coordinates longitude $\lambda \in [-\pi, \pi]$ and latitude $\phi \in [-\pi/2, \pi/2]$ into a higher-dimensional encoding space. A small neural network (NN) then projects the encoding to a typically lower-dimensional output y, such as a classification logit.

### A.2.1  NEURAL NETWORKS

**LINEAR** simply consists of a linear layer on top of the positional embedding. It serves as a baseline to ascertain the quality of the representation produced by the positional embedding alone.

**FCNET** is a multi-layer perceptron with dropout (Srivastava et al., 2014) and residual connections between layers (He et al., 2016), as illustrated in fig. 1a. Proposed by Mac Aodha et al. (2019), it is the default architecture used by the follow-up work.

**SIRENNET**, the sinusoidal representation network, is a recent architecture proposed by Sitzmann et al. (2020) to represent complex natural signals with the use of periodic activation functions. As shown in fig. 1b, a layer of SIREN comprises a linear projection, dropout, and the sine activation function. It is the first time that this network is used for location encoding.

### A.2.2  POSITIONAL EMBEDDINGS

**DIRECT**, i.e., identity, serves as an often-used baseline

$$\text{PE}(\lambda, \phi) = (\lambda, \phi) \quad (10)$$

**CARTESIAN3D** encodes the spherical $\lambda, \phi$ into 3D-Cartesian coordinates $x, y, z$ is an popular alternative Tseng et al. (2023; 2022)

$$
\text{PE}(\lambda, \phi) = \left[ \overbrace{\cos\phi\cos\lambda}^{x}, \overbrace{\cos\phi\sin\lambda}^{y}, \overbrace{\sin\phi}^{z} \right] \tag{11}
$$

**WRAP** (Mac Aodha et al., 2019) embeds the spherical coordinates $(\lambda, \phi)$ into sine and cosine functions to avoid discontinuities on the dateline:

$$
\text{PE}(\lambda, \phi) = [\cos\lambda, \sin\lambda, \cos\phi, \sin\phi] \tag{12}
$$

**GRID** (Mai et al., 2020b) enhances WRAP by adding multiple frequencies $\lambda_s = \frac{\lambda}{\alpha_s}$ and $\phi_s = \frac{\phi}{\alpha_s}$ with a scale-dependent factor $\alpha_s = r_{\min} \cdot \left( \frac{r_{\max}}{r_{\min}} \right)^{\frac{1}{S-1}}$ that allows users to specify the minimum and maximum radii $r_{\min}, r_{\max}$ that can be resolved at multiple scales up to $S$:

$$
\text{PE}(\lambda, \phi) = \bigcup_{s=0}^{S-1} [\cos\lambda_s, \sin\lambda_s, \cos\phi_s, \sin\phi_s] \tag{13}
$$

**THEORY** Mai et al. (2020b) extends GRID by projecting the locally-rectangular $\lambda, \phi$ coordinates into a hexagonal grid with three 120°-spaced basis functions $\mathbf{a}_1 = [1,0]^\top$, $\mathbf{a}_2 = [-\frac{1}{2}, \frac{\sqrt{3}}{2}]^\top$, and $\mathbf{a}_3 = [-\frac{1}{2}, -\frac{\sqrt{3}}{2}]^\top$:

$$
\text{PE}(\lambda, \phi) = \bigcup_{s=0}^{S-1} \bigcup_{i=1}^{3} \left[ \cos\left( \frac{\langle [\lambda, \phi], \mathbf{a}_i \rangle}{\alpha_s} \right), \sin\left( \frac{\langle [\lambda, \phi], \mathbf{a}_i \rangle}{\alpha_s} \right) \right] \tag{14}
$$

**SPHEREC** Mai et al. (2023b) is a simplified version of the Double Fourier Sphere (DFS) (Merilees, 1973; Orszag, 1974):

$$
\text{PE}(\lambda, \phi) = \bigcup_{s=0}^{S-1} [\sin\phi_s, \cos\phi_s\cos\lambda_s, \cos\phi_s\sin\lambda_s] \tag{15}
$$

**SPHEREC+** (Mai et al., 2023b) combines SPHEREC and GRID:

$$
\text{PE}(\lambda, \phi) = \bigcup_{s=0}^{S-1} [\sin\phi_s, \cos\phi_s\cos\lambda_s, \cos\phi_s\sin\lambda_s] \cup \tag{16}
$$

$$
\bigcup_{s=0}^{S-1} [\cos\lambda_s, \sin\lambda_s, \cos\phi_s, \sin\phi_s] \tag{17}
$$

**SPHEREM** (Mai et al., 2023b) is a simplified version of the Double Fourier Sphere (DFS) (Merilees, 1973; Orszag, 1974):

$$
\text{PE}(\lambda, \phi) = \bigcup_{s=0}^{S-1} [\sin\phi_s, \cos\phi_s\cos\lambda, \cos\phi\cos\lambda_s, \cos\phi_s\sin\lambda, \cos\phi\sin\lambda_s] \tag{18}
$$

**SPHEREM+** (Mai et al., 2023b) combines SPHEREM and GRID:

$$
\text{PE}(\lambda, \phi) = \bigcup_{s=0}^{S-1} [\sin\phi_s, \cos\phi_s\cos\lambda, \cos\phi\cos\lambda_s, \cos\phi_s\sin\lambda, \cos\phi\sin\lambda_s] \cup \tag{19}
$$

$$
\bigcup_{s=0}^{S-1} [\cos\lambda_s, \sin\lambda_s, \cos\phi_s, \sin\phi_s] \tag{20}
$$

**SPHERICAL HARMONICS** is detailed in section 2.2.

