# OpenReview forum: "Geographic Location Encoding with Spherical Harmonics and Sinusoidal Representation Networks"
_ICLR.cc/2024/Conference — ICLR 2024 spotlight_

### Official Review · Reviewer_YTje · 2023-10-30

**Soundness:** 4 excellent
**Presentation:** 3 good
**Contribution:** 3 good
**Rating:** 5
**Confidence:** 4

**Summary:**

A method to learn features representations of geographical data observed on the Earth's surface is presented.  The method is based on intrinsic neural fields, using spherical harmonic functions for the positional encoding, combined with SIRENs which adopt sine non-linearities as periodic activation functions (the spherical harmonic embedding can of course also be combined with other neural networks).  It is also shown that SIRENs can be see as a form of learned Double Fourier Sphere (DFS) positional embedding.  A number of experiments are presented, across synthetic and real-world problems, demonstrating the effectiveness of the representation for machine learning tasks, typcially classification.  The proposed method demonstrates an improvement in performance over existing techniques.

**Strengths:**

The method is straightforward and flexible since it simply involves a spherical harmonic encoding coupled with a neural network (e.g. fully-connected, SIREN).  The poles on the sphere can be handled directly since the continuous spherical harmonic functions can be evalued for any coordinates, including the poles, whereas alternative approaches based on equirectangular sampling of the sphere often have large numbers of samples near the poles, which can induce artefacts.

**Weaknesses:**

Positional encoding for implicit neural representations on manifolds, such as the sphere, have been considered previously in Grattarola & Vandergheynst (2022), which is cited, and also in [Koestler et al. (2022)](https://arxiv.org/abs/2203.07967), which is not cited but should be.  Grattarola & Vandergheynst do not specifically consider the sphere and spherical harmonics, instead focusing on graph representations, but consider an emedding based on the eigenfunctions of the graph Laplacian.  Koestler et al. consider general manifolds, with an embedding based on the eignenfunctions of the Laplace-Beltrami operator.  On the sphere, these eignerfunctions are specifically the spherical harmonics.  So an embedding based on spherical harmonics as presented in this article is not new, although in this work the embedding is combined with SIRENs and extensive experiments on the sphere are performed.

One limitation of the proposed approach is that it is limited to very low degrees L on the sphere, typically L of 20 or at most 40.  The spherical harmonics are precomputed analytically, which while fine for these very low degrees will not scale to higher degrees.

It is indeed the case that combining the spherical harmonic encoding with SIRENS typically gives the best performance (see Table 1).  However, in many cases the improvement is not that substantial, compared to the next-best method.  The improvements for the ERA 5 dataset appear to be more marked.

**Questions:**

- Could the authors elaborate how their work fits into the context of prior work by Grattarola & Vandergheynst (2022) and [Koestler et al. (2022)](https://arxiv.org/abs/2203.07967).

- It seems combined spherical harmonic encoding with SIRENs is generally best.  However, for the ERA 5 dataset it seems a fully connected network was significantly better than combing with SIREN.  Do the authors have any idea why a fully connected network was best here?

- Do the authors have any thoughts on how they could extend the method to higher degrees L?

---

> ### Author Response · Authors · 2023-11-16
> **Response by the Authors**
>
> Thank you very much for the detailed and critical evaluation of our paper and the precise questions!
>
> ### Response to Weaknesses
>
> > Reviewer:
> > Positional encoding for [...] manifolds, such as the sphere, have been considered previously in [...] Koestler et al. (2022), which is not cited but should be.
>
> Thank you for the valuable insights in Koestler et al. (2022). We modified the introduction to acknowledge this work (blue font).
>
> > Reviewer:
> > Limitation to low degrees L.
>
> While the computational complexity increases with higher L, as shown in Figure 5, we would like to highlight that the computational runtime is __not excessive even until L=100__, which is the largest L harmonic function we pre-computed. Pre-computing the analytic functions of higher L is easily possible, but we found that such a high L was not necessary for the experiments of this paper.
>
> To illustrate that runtime is not a major limitation, a forward pass of 256 coordinates up to L=100 (in total L$^2$ harmonics) takes only __716 ms ± 17.7 ms__ on a Macbook CPU.
> In comparison, a forward pass through a ResNet 50 vision model on a [256 x 3 x 224 x 224] image batch takes 5.33 s ± 225 ms on the same computer. Hence, a SirenNet(SH) location encoder (even with a high L) will likely still contribute less to the runtime than a common vision model.
>
> > Reviewer:
> > It is indeed the case that combining the spherical harmonic encoding with SIRENS typically gives the best performance (see Table 1). However, in many cases the improvement is not that substantial.
>
> While the performance gain of SH may seem comparatively small, it is generally _consistent across all datasets_ and _across neural networks_. We show in Fig 4b) that SHs are most effective at high latitudes and polar areas where the spherical geometry of our planet matters the most. Here, other DFS-based location encoders systematically lose accuracy since their underlying assumptions on the geometry (i.e., the rectangular domain of lon/lat) are not met. Since only a few test points are located at high latitudes or polar areas, the accuracy benefits are numerically small in the tables.
>
> ### Response to Questions
>
> > Reviewer:
> > Could the authors elaborate on how their work fits into the context of prior work by Grattarola & Vandergheynst (2022) and Koestler et al. (2022).
>
> We see our work targeted towards the concrete modality of geographic location encoding in lon/lat coordinates on our spherical Earth. This is reflected by the tested datasets (land-ocean classification, ERA-5 climate science, iNaturalist species distribution modeling).
> We understand this work as a special case of learning implicit neural representations on the Sphere as specific non-euclidean geometry. We highly appreciate the fundamental works of Koestler et al. (2022) and Grattarola & Vandergheynst (2022). The generality of these papers to learn functions on arbitrary geometries through the expression as eigenvectors in graphs is highly valued. In this work, we only focus on the sphere of our Planet. Hence, we can focus on purpose-built, efficient implementations of Spherical Harmonics with analytically pre-defined functions of certain degrees and orders.
>
> > Reviewer:
> > It seems combined spherical harmonic encoding with SIRENs is generally best. However, for the ERA 5 dataset it seems a fully connected network was significantly better than combining with SIREN. Do the authors have any idea why a fully connected network was best here?
>
> FcNet is a well-tested network for location encoding. In our opinion, FcNet remains a reasonable choice when it is paired with a suitable positional encoding. SirenNet achieves _similar and often better results_ and, crucially, _SirenNet is less sensitive to the underlying positional encoding_ and _predicts reasonably well even with direct encoding_ of longitude and latitude. In Section 2.3, we have shown that a specifically initialized SirenNet layer becomes equivalent to the Grid DFS-embedding. Out of these reasons, we see SirenNet as an architecturally simpler replacement over FcNet that archives reliably good results irrespectively of the positional embedding.
>
> > Reviewer:
> > Do the authors have any thoughts on how they could extend the method to higher degrees L?
>
> One heuristic to enable higher degrees L would be to pre-compute the numerical values of harmonics of explicit training points with a high L (like L=1000). These harmonics can then be loaded directly from the disk.
>
> A more principled solution is likely found in Geodesy, where the Earth's gravity field is stored as weighted spherical harmonics up to several hundred Legendre polynomials (i.e., Linear(SH)). In this field, [Jekeli et al., 2006](https://link.springer.com/article/10.1007/s00190-006-0123-z) and [Balmino et al., 2012](https://link.springer.com/article/10.1007/s00190-011-0533-4) investigated approximations that allow integrating signals with very large L. We plan to explore this direction in future collaborative work.

---

> > ### Comment · Reviewer_YTje · 2023-11-23
> >
> > Thank you for the comments and clarifications.

---

### Official Review · Reviewer_g3zK · 2023-11-01

**Soundness:** 3 good
**Presentation:** 3 good
**Contribution:** 3 good
**Rating:** 8
**Confidence:** 3

**Summary:**

This paper proposes a location encoder which combines spherical harmonics (SH) coordinate embeddings with sinusoidal representation networks (SirenNets) and argues that SH encodings enable accounting better for the geometry of the Earth than existing methods using Double Fourier Sphere which project coordinates to rectangular domain.

Several positional embedding methods in combination with different neural networks to obtain location encoders are compared across 4 tasks. They claim that SH with SirenNets are an effective way of encoding geographical location, in particular for tasks where there is data in polar regions or at a global scale.
They also find that SirenNets perform competitively on their own (without any encoding of the latitude and longitude) which they explain by showing that SirenNets can written out as DFS embeddings.

**Strengths:**

- The paper is clear and well-written.
- While neither SH nor SirenNets are novel, their combination appears as a well-motivated approach to location encoding, and has never been used in this context.
- The proposed method attempts at addressing challenges around the poles, an issue which seems to have been overlooked with methods assuming a rectangular data space.
- The results obtained on the 4 datasets suggest that the proposed combination of SH and SirenNet is an effective way of encoding location.

**Weaknesses:**

- While we acknowledge the work put into designing tasks to showcase challenges specific to geographic location encoding, it would have been interesting to see more comparisons on datasets that have been used by previous work on this topic, besides the iNaturalist task (e.g. the fMoW dataset used in the Sphere2Vec paper cited in this work).
- The task on the ERA5 dataset was designed specifically for this work. It is difficult to assess whether the gain in using SH would also be significant in real-life climate science tasks, which is a domain of application that the paper puts forward. Would it be possible to integrate this method other tasks more common with ERA5 such as downscaling?

**Questions:**

- I am curious to hear the authors' intuition about the combination of SirenNet and some DFS based encodings performing worse than SirenNet alone? (e.g. SphereC+ + SirenNet on INaturalist 2018)
- I may have missed this information but what is the number of Legendre Polynomials in the experiments results reported in the tables? I would also be curious to have the performance of the different models of Figure 5 with the number of polynomials alongside the computational efficiency. As in, is the extra computational cost associated with using SH worth bearing with a higher number of polynomials in terms of gain in performance for L > 20? It seems that not (from looking at Fig4a) but it would be helpful to compare the performance of the different location encoding methods.
-  I am surprised that SphereM performs so poorly in comparison to the other positional embeddings (the difference in performance is not as striking in the Sphere2vec paper on their tasks). What is your take on this poor performance?

---

> ### Author Response · Authors · 2023-11-16
> **Response by the Authors**
>
> Thank you for the detailed evaluation, precise questions, and insightful comments!
>
> ### Response to the Weaknesses
>
> > Reviewer:
> > It would have been interesting to see more comparisons on datasets [...], e.g., the fMoW dataset.
>
> We considered including FMoW initially but decided against it, as the accuracy benefit of adding coordinate information is only 1.62 % (see Mai et al., 2023; Sphere2Vec Table 3; no std devs given). We believe that this margin is too small to show any systematic differences between the location encoders. Instead, we focused rather on iNaturalist, where the benefit of adding geolocation adds a 12.3% improvement (Table 1c) over the image-only model.
>
> > Reviewer:
> > Would it be possible to integrate this method other tasks more common with ERA5 such as downscaling?
>
> The location encoder is not restricted to gridded data and can predict ERA5 variables continuously. Hence, generating a 1km rather than 30km grid (as in downscaling) is possible on a technical level. Still, adding further (higher-res.) covariates, like detailed topographic features, is likely necessary to improve the effective resolution. The location encoder can be used here as a module for a larger downscaling model.
>
> ### Response to Questions
>
> > Reviewer:
> > Intuition about the combination of SirenNet and some DFS based encodings performing worse than SirenNet alone?
>
> Sphere{C,M} keep one lat/lon coordinate at the original (lowest-frequency) scale while scaling up the other. We hypothize that the lowest scale may be too low-resolution, which can be potentially detrimental. Beyond that, we have no further intuition other than that DFS-embeddings are not optimal for the Sphere in general (violated assumption of rectangular lat/lon domain). Regarding the good performance of SirenNet alone (i.e., with Direct), we show in Section 2.3 that a single Siren layer with specifically set weights can produce the same embedding as Grid. Hence, it is not surprising that Siren with direct coordinates performs competitively well.
>
> > Reviewer:
> > I may have missed this information but what is the number of Legendre Polynomials in the experiments results reported in the tables?
>
> We performed hyperparameter tuning separately for each NN-PE combination, as described in the Appendix Section "A.1 Implementation details and hyperparameter tuning".
>
> Concretely, every model in the results tables has a different configuration for the SH models. We tune the Legendre polynomials L between 10 and 30 in steps of 5 polynomials. In the best hyperparameter file, we find the entire value range (10 to 30) present. Indeed, L controls the smoothness of the interpolation directly. A too-low L may be too low-frequency to capture the signal, while a too-high L may lead to overfitting around particular training points and not considering nearby test points.
>
> > Reviewer:
> > I would also be curious to have the performance of the different models of Figure 5 with the number of polynomials alongside the computational efficiency.
>
> Thank you for this suggestion. We provide the accuracies of the models below. Note that SphereC+ does not have an L parameter to control the degree of interpolation, but an $r_{min}$ parameter that determines the minimum radius. We compare both using the maximum resolvably frequency formula $r_{min} = f_{max} = \frac{180°}{2L}$.
>
> You can find the accuracies of the runtime experiment for Figure 5 below:
>
> |__Spherical Harmonics__|L=2|L=5|L=10|L=20|L=40|
> |-|-|-|-|-|-|
> | Siren(SH) analytic|7.74 %|36.00 %|82.03 %|95.89 %|94.63 %|
> |__SphereC+__|$r_{min}=90°$|$r_{min}=36°$|$r_{min}=18°$|$r_{min}=9°$|$r_{min}=4°$|
> |Siren(SphereC+)|13.44 %|15.81 %|20.55 %|34.71 %|49.87 %|
>
> Note that SphereC+ is not as accurate as in the comparison tables, as $r_{min}=4°$ is likely too restrictive (many, but not all, optimal $r_{min}$ hyperparameters are 1). We chose $r_{min}$ to be comparable to L rather than optimal with respect to predictive accuracy (as in Tables 1,2). Accuracy-wise, Siren(SH) with L=20 is slightly more accurate than Siren(SH) L=40. Hence, a higher L does not automatically lead to better accuracies.
>
> > Reviewer:
> > What is your take on SphereM poor performance?
>
> Thank you for noticing that.
> We double-checked our implementation of SphereM{+}, which corresponds exactly to the code provided by Mai et al., 2023.
>
> To make sure, we also re-ran the land-ocean classification on a different computer (now Macbook Pro on CPU/originally a GPU workstation), which resulted in very similar results to Table 1b:
>
> |PE, NN|Linear|FcNet|Siren|
> |-|-|-|-|
> |SphereC|$79.9 \pm 0.3$|$94.9 \pm 0.1$|$95.2 \pm 0.1$|
> |SphereC+|$84.4 \pm 0.0$|$95.1 \pm 0.1$|$95.4 \pm 0.0$|
> |SphereM|$74.0 \pm 0.0$|$89.0 \pm 0.3$|$88.3 \pm 0.4$|
> |SphereM+|$81.9 \pm 0.2$|$91.6 \pm 0.2$|$93.7 \pm 0.1$|
> |SH (ours)|$94.1 \pm 0.0$|$95.7 \pm 0.1$|$95.8 \pm 0.1$|
>
> Hence, we do not have an explanation for the poor SphereM performance and can only stress that we run all models in the same environment.

---

> > ### Comment · Reviewer_g3zK · 2023-11-22
> >
> > Thank you for your responses to my and the other reviewers' questions!
> > They address my concerns and I will raise my rating.

---

### Official Review · Reviewer_BVXQ · 2023-11-04

**Soundness:** 3 good
**Presentation:** 3 good
**Contribution:** 3 good
**Rating:** 8
**Confidence:** 3

**Summary:**

In this paper, authors propose a ML model for learning feature representation for geographical space, based on spherical harmonic basis functions as positional embeddings that are well-defined across the entire globe, including the poles. Specifically, author propose ropose Spherical Harmonic (SH) coordinate embeddings that work well with a few tested neural networks, especially well when paired with Sinusoidal Representation Networks (SirenNet) .

Authors test the proposed embeddings with a few state-of-the-art models and demonstrate improved performance with their method in comparison to previous research.

**Strengths:**

The paper is well written, the experiments include some key datasets, used in climate science (such as ERA5). the results are well-presented and sound. The paper meets all the requirements for the ICLR publication.

**Weaknesses:**

The paper is quite dense, with prevalence of abbreviations and mathematical notation over the description, which makes it difficult to read, especially in  sec.3 Datasets and experiments. Readers without the knowledge of the specific datasets won't be able to understand the application. I would suggest reduce the number of the use-cases to 3 (g.g ERA5 and iNaturalist) and describe in details what exactly did authors do.

**Questions:**

Please see the previous section: part 3 is too dense, please insert more explanations on what was done and descriptions. Alternatively, you can include mere details as appendices for better paper understanding

---

> ### Author Response · Authors · 2023-11-15
> **Response by the Authors**
>
> Thank you for the precise and accurate analysis of our work!
>
> > Reviewer:
> > The paper is quite dense, with prevalence of abbreviations and mathematical notation over the description, which makes it difficult to read, especially in sec.3 Datasets and experiments. Readers without the knowledge of the specific datasets won't be able to understand the application. I would suggest reduce the number of the use-cases to 3 (g.g ERA5 and iNaturalist) and describe in details what exactly did authors do.
>
> We greatly appreciate your comments regarding the density and details and have taken your suggestion by extending the dataset section 3 in the revised version by adding additional sentences marked in blue in the revised paper.
>
> We would still favor keeping all four use-cases/datasets in the main paper, as every dataset reflects a different aspect of performance. We did our best to line out the major goals and findings of our study with a clear overall structure and text.
>
> Thank you again for the accurate summary of for paper and we remain at your disposal if any further questions remain.

---

### Official Review · Reviewer_K9PX · 2023-11-10

**Soundness:** 2 fair
**Presentation:** 4 excellent
**Contribution:** 3 good
**Rating:** 6
**Confidence:** 4

**Summary:**

This work considers the problem of encoding geographical locations to facilitate downstream geospatial prediction tasks. The paper proposes a new approach (spherical harmonics / SH) to geographical location encoding and compares it to other approaches on a collection of real and synthetic downstream tasks. A key focus of the paper is separately evaluating the contributions of the location encoding and the network into which the location encoding is passed.

Generally, I think this paper has value but there are a few concerns about the experimental results that need to be cleared up.

EDIT: Increase from marginally below to marginally above after discussion.

# References

@inproceedings{martinez2017simple,
  title={A simple yet effective baseline for 3d human pose estimation},
  author={Martinez, Julieta and Hossain, Rayat and Romero, Javier and Little, James J},
  booktitle={Proceedings of the IEEE international conference on computer vision},
  pages={2640--2649},
  year={2017}
}

@inproceedings{mac2019presence,
  title={Presence-only geographical priors for fine-grained image classification},
  author={Mac Aodha, Oisin and Cole, Elijah and Perona, Pietro},
  booktitle={Proceedings of the IEEE/CVF International Conference on Computer Vision},
  pages={9596--9606},
  year={2019}
}

**Strengths:**

* The paper is generally well-written, with few typos or unclear passages.
* The problem under consideration is important and timely.
* Results are presented with error bars.
* The figures are well-made and informative.
* The evaluation tasks are reasonable.
* The detailed comparison of different combinations of location encodings and network is useful for the community - as the authors point out, these have not often been studied as separate modules.
* The performance improvements (with a linear encoder) are impressive.
* Solid hyperparameter tuning procedures.

**Weaknesses:**

# When are the performance gains from SH worth it?

When using FCNet or SirenNet (which is not much of a burden in practice), the gains from SH are fairly small (<1%). The gains from SH are only large when using a low-capacity network. Similarly, in Figure 4(a) we see that the difference between $L=10$ and $L=20$ is only significant when using low-capacity networks. To me, this seems to suggest that "expressiveness" is the main issue. From a user's perspective, does it matter whether the expressiveness comes from the input encoding or the network? What's the difference in computational efficiency between SH + SirenNet and X + SirenNet where X is another high-performing input encoding? What's the best argument (combining computational efficiency and performance) that SH is worthwhile to use as a component in a real system compared to the alternatives?

# What is the trade-off between performance and efficiency?

Generally speaking, the "take home" efficiencies of the different methods are not clear. Figure 5 shows the computation time required to compute SH under different implementations, but the important thing is the trade-off between performance and computational efficiency. E.g. is SH + Linear faster than SH + FCNet? This would be very helpful for evaluating the practical usefulness of SH.

# Isn't network capacity and degree of fit a significant confounder in the experiments?

Following on from the previous point, FCNet and SirenNet are "typically" implemented with a certain number of layers. I might have missed it, but how many layers does SirenNet have in this work? Are FCNet and SirenNet matched in this sense? Do they behave differently as the capacity of the networks increases? What is the effect of training duration / how well the network has fit to the training data? These factors seem very important for comparing the results for these two networks, but I do not see them discussed in the text. This makes it difficult to evaluate key claims in the paper.

**Questions:**

See the questions/headings in "weaknesses".

# Misc. Comments
* I suggest underlining the second-best value in each column in Table 1 to make it easier for the reader to see the magnitude of the change between the best and second-best method.
* For context, I believe that "FCNet" in [mac2019presence] was based on an architecture found in earlier work in pose estimation [martinez2017simple].
* It's not clear that the set union notation in Eq. (2) is being used to build a vector - please clarify in the text.
* Is there any intuition for why DFS don't perform well empirically?
* I think the reader would benefit from a discussion of how the test sets were chosen, and why they are appropriate in a spatial prediction context.

---

> ### Author Response · Authors · 2023-11-15
> **Response by the Authors**
>
> Thank you very much for your careful and critical reading of our work.
>
> ### Response to Main Questions
>
> > Reviewer:
> > When are the performance gains from SH worth it?
>
> The quantitative differences between SH-based encoders and DFS-based encoders may be numerically small but are systematic for a) different neural networks and b) present across different datasets (Tables 1 and 2).
>
> Spherical harmonics are inherently better suited for spheres than DFS-based encodings that assume a rectangular domain between longitude and latitude. As shown in Figure 4b), this assumption does not hold at high latitudes, where DFS-based encoders drop systematically in accuracy, while SH remains accurate.
>
> > Reviewer:
> > The gains from SH are only large when using a low-capacity network. [...]. To me, this seems to suggest that "expressiveness" is the main issue.
>
> An expressive high-capacity neural network, such as FcNet or Siren, can compensate for inaccuracies of sub-optimal DFS-based embeddings.
> In our view, the good performance of SH even on the Linear "low-capacity network" highlights that spherical harmonic basis functions alone are inherently better suited for global-scale prediction tasks.
>
> > Reviewer:
> > What's the difference in computational efficiency between SH + SirenNet and X + SirenNet where X is another high-performing input encoding?
>
> The table below shows runtimes for the land-ocean dataset (Table 1b) $\pm$ from 5 runs. We reran the models on CPU on a Macbook. The SH embeddings in this used 10 Legendre Polynomials (determined by hyperparameter tuning as detailed in the appendix).
>
> |runtime in sec|Siren(SH) L=10|Siren(SphereC)|Siren(SphereC+)|Siren(SphereM)|Siren(SphereM+)|
> |:-|-:|-:|-:|-:|-:|
> |train|23.31 $\pm$ 2.14|19.23 $\pm$ 1.37| 12.84 $\pm$ 1.23|14.00 $\pm$ 1.57|43.12 $\pm$ 7.96|
> |test|0.45 $\pm$ 0.03|0.80 $\pm$ 0.03|0.45 $\pm$ 0.00|0.44  $\pm$ 0.04|0.70 $\pm$ 0.04|
>
> In practice, we see no disadvantage in terms of the runtime of one over another.
>
> > Reviewer:
> > What's the best argument [...] that SH is worthwhile to use as a component in a real system compared to the alternatives?
>
> SH basis functions are effective for all networks and perform well even for simple (i.e., Linear) neural networks (Tables 1,2), they can be faster than DFS encodings until L=20 (Figure 5) and are even up to L=100 not excessive (5.62 sec. run time for 2000 points on Macbook CPU).
>
> > Reviewer:
> > What is the trade-off between performance and efficiency? [...] The important thing is the trade-off between performance and computational efficiency. E.g. is SH + Linear faster than SH + FCNet?
>
> The runtime difference between Linear(SH) and FCNet(SH) (with L=15 polynomials) of 2k points on the land-ocean dataset CPU/Macbook is minor: Linear(SH) took 0.91 s, while FCNet(SH) took 0.97 s.
>
> Hence, Linear(SH) is slightly faster than FCNet(SH) by 0.06 s due to the more complex FCNet model.
>
> Below, we also provide the accuracy % and runtimes in seconds from Table 5 (inference of 10000 test points of the checkerboard dataset on GPU):
> |Siren(SH) analytic|L=2|L=5|L=10|L=20|L=40|
> |-|-|-|-|-|-|
> |accuracy in %|7.74 %|36.00 %|82.03 %|95.89 %|94.63 %|
> |runtime in s|0.06 s|0.07 s|0.09 s|0.19 s|0.99 s|
>
> A larger L (and longer compute runtime) does not automatically lead to better results (i.e., here L=20 is better than L=40).
>
> > Reviewer:
> > Isn't network capacity and degree of fit a significant confounder in the experiments? [...] How many layers does SirenNet have in this work?
>
> We tune each PE-NN combination separately for 100 epochs. The hyperparameter space includes hidden dimensions within the networks, number of layers, learning rate, weight decay, number of frequencies S, $r_{min}$. The exact parameter range is detailed in the appendix.
>
> > Reviewer:
> > Are FCNet and SirenNet matched in this sense?
>
> In the hyperparameter tuning procedure, they are matched in the sense that we take the best-performing model configuration on the validation set.
>
> > Reviewer:
> >Do they behave differently as the capacity of the networks increases?
>
> We see no clear connection between model capacity and classification/interpolation performance. Simpler (lower-capacity) models also interpolate better towards spatially different test points.
>
> > Reviewer:
> > What is the effect of training duration / how well the network has fit to the training data?
>
> During training, we monitor the validation loss and perform early stopping with patience of 30 epochs. We do this for all NN-PE combinations consistently, as detailed in the revised appendix.
>
> ### Misc Comments
>
> We integrated the misc comments in the revised manuscript and appendix (blue font).

---

> ### Comment · Reviewer_K9PX · 2023-11-22
> **Response to authors**
>
> Thank you for the thoughtful responses!
>
> It seems to be the case that SH is neither much faster nor much more accurate than the alternatives considered in the paper. (Please correct me if I've misunderstood.) This is not a problem in and of itself (since the paper has other merits), but it **is** a problem if the paper fails to make this clear to the reader. I would strongly encourage revisions for clarity along these lines.
>
> Regardless, I think the paper has value and will be of interest to the community, so I will increase my rating to weak accept. Among other things, the careful separate analysis of encodings and networks is a great contribution.

---

> > ### Author Response · Authors · 2023-11-22
> > **Thank you for the response & follow-up on being only slightly more accurate**
> >
> > We thank the reviewer for opening the discussion, thoroughly reading our responses, and raising the score!
> >
> > We would like to clarify regarding not being _much_ more accurate than other approaches.
> >
> > We agree that Siren(SH) is _on global average_ (Tables 1,2) consistently but numerically _only slightly_ more accurate than other location encoders used in related works, i.e., FcNet(Grid), FcNet(Wrap).
> > However, we highlight at higher latitudes, the difference is much larger (up to 10%-FcNet(Grid) above 50 degrees latitude N/S), as we show quantitatively in Fig 4b). At these latitudes, the underlying assumption of DFS encodings, i.e., rectangular lon/lat domain, is violated more severely. In Figure 4c) we also show artifacts at the poles that can appear with DFS embeddings like Grid or SphereM+.
> >
> > Thank you for raising this point. We will revise the text to make it more clear.

---

> > > ### Comment · Reviewer_K9PX · 2023-11-22
> > > **Thank you for the clarification**
> > >
> > > Yes, that's a fair point - thank you for clarifying.

---

### Author Response · Authors · 2023-11-15
**General Response by the Authors to all Reviewers**

We thank all reviewers for their thorough and careful reading of our paper and the accurate summaries throughout.

We were delighted to read that the paper "has value" (R-K9PX) and that the manuscript has been deemed "clear" (R-g3zK) and "well written" (R-K9PX, R-BVXQ, R-g3zK) and that "the problem is important and timely" (R-K9PX). While we build on existing concepts like Spherical Harmonics and SirenNet, it has been acknowledged that their combination has never been used in this context and is well-motivated (R-g3zK).
We stress the applicability of our "straightforward and flexible method" (R-YTje) that we tested on "key datasets used in climate science (such as ERA5)" (R-BVXQ). And we are glad to hear that our "results are sound" (R-BVXQ) and our hyperparameter tuning procedures were solid (R-K9PX)

We are confident that we can clear up all remaining concerns in our individual responses to each reviewer throughout the discussion period. We uploaded a revised version of the paper and appendix and marked changes in blue font. We remain at your disposal if any open questions remain.

---

### Meta-Review · Area_Chair_huZU · 2023-12-09

**Metareview:**

A large number of tasks involve geographically distributed data, and there has recently been interest in incorporating positional information into the solution of such problems. This paper considers how to improve the encoding of geographical location with neural network architectures based on spherical harmonics, in particular in combination with sinusoidal representation networks, finding improved results across a range of tasks. The reviewers largely agree this is a valuable and convincing contribution, and it is an area of great topicality within the remote sensing + ML community. Accordingly, I recommend a spotlight presentation.

**Justification For Why Not Higher Score:**

Given the concerns in K9PX's review in particular, I lean towards spotlight rather than oral, but depending on number of orals it could go that way instead.

**Justification For Why Not Lower Score:**

The reviewers largely agree this is a valuable and convincing contribution, and it is an area of great interest at the moment within the remote sensing + ML community.

---

### Decision · Program_Chairs · 2024-01-16

Accept (spotlight)